# Carbohydrate Considerations for Athletes with a Spinal Cord Injury

**DOI:** 10.3390/nu13072177

**Published:** 2021-06-24

**Authors:** Belinda Ruettimann, Claudio Perret, Jill A. Parnell, Joelle Leonie Flueck

**Affiliations:** 1Department of Health Sciences and Technology, ETH Zurich (Swiss Federal Institute of Technology Zurich), CH-8003 Zurich, Switzerland; belinda.ruettimann@sportmedizin-nottwil.ch; 2Swiss Paraplegic Centre Nottwil, Institute of Sports Medicine, CH-6207 Nottwil, Switzerland; claudio.perret@sportmedizin-nottwil.ch; 3Department of Health and Physical Education, Mount Royal University, 4825 Mount Royal Gate SW, Calgary, AB T3E 6K6, Canada; jparnell@mtroyal.ca

**Keywords:** sports nutrition, paraplegia, tetraplegia, Paralympic, wheelchair athlete, exercise, recommendations

## Abstract

The Paralympic movement is growing in popularity, resulting in increased numbers of athletes with a spinal cord injury (SCI) competing in various sport disciplines. Athletes with an SCI require specialized recommendations to promote health and to maximize performance, as evidenced by their metabolic and physiological adaptations. Nutrition is a key factor for optimal performance; however, scientifically supported nutritional recommendations are limited. This review summarizes the current knowledge regarding the importance of carbohydrates (CHO) for health and performance in athletes with an SCI. Factors possibly affecting CHO needs, such as muscle atrophy, reduced energy expenditure, and secondary complications are analyzed comprehensively. Furthermore, a model calculation for CHO requirements during an endurance event is provided. Along with assessing the effectiveness of CHO supplementation in the athletic population with SCI, the evaluation of their CHO intake from the available research supplies background to current practices. Finally, future directions are identified. In conclusion, the direct transfer of CHO guidelines from able-bodied (AB) athletes to athletes with an SCI does not seem to be reasonable. Based on the critical role of CHOs in exercise performance, establishing recommendations for athletes with an SCI should be the overall objective for prospective research.

## 1. Introduction

Sport for individuals with a spinal cord injury (SCI) has undergone a remarkable development in recent decades, with improvements in performance and participation [1]. While primarily focusing on rehabilitation in the early days, the possibilities to engage in high-performance athletics grew progressively over time. The Paralympic Games now represent the quadrennial highlight for elite-level, Para-athletes and include hundreds of competitors and mainstream media exposure [1,2,3]. An inspiring effect is attributed to the Paralympic Games, especially for children and young people with disabilities, regarding self-perception and confidence to participate in sport [2]. Furthermore, the annual incidence of an SCI is estimated to be between 250,000 and 500,000 individuals worldwide [4]. Considering the aforementioned popularity of the Paralympic Games and the relatively high incidence of SCIs, it seems natural to assume a continually increasing number of individuals with an SCI exercising in various types of sport. Accordingly, competition at both the elite and amateur level will further intensify.

Athletes with an SCI are committed and dedicated to their athletic pursuits and deserve support and guidelines comparable to those available for able-bodied (AB) athletes [1]. However, there is only sparse literature available in the athletic population, with SCI clamoring for research focused on the unique needs of these competitors. In particular, there is a need for evidence-based nutritional recommendations, since nutrition is a key factor for peak performance [1,5]. Owing to insufficient research and guidelines tailored to individuals with an SCI, the standards for nutritional requirements and recommendations applied to athletes with an SCI are presently based on those for AB athletes [1]. Yet, athletes with an SCI represent a unique subgroup of the highly diverse Paralympic community that likely requires specialized recommendations to promote health and optimize performance [1]. A fundamental understanding of the factors affecting athletes with an SCI (e.g., level of injury, substrate utilization and secondary complications) is essential in modifying the nutritional recommendations [1]. Specifically, carbohydrate (CHO) recommendations are of great interest, as they are essential to performance capacity and training adaptations [5,6]. CHOs are a key fuel for the brain and central nervous system and represent a flexible substrate for muscular work [5]. Based on its utilization in both the aerobic and anaerobic pathways, CHO is supportive over a large range of exercise intensities [7]. Furthermore, there is significant evidence that the performance of longer lasting events is enhanced by maintaining high CHO availability [5,6,8]. Inadequate and/or depleted CHO stores are associated with reduced exercise capacity, impaired motor skill, decreased concentration, and increased perception of effort, as well as premature fatigue [5]. Finally, recent work has identified that glycogen not only plays an important role as a muscle substrate, but is also essential in regulating the muscle’s adaptation to training [5,9,10]. While the absolute size of body CHO stores is limited, they are susceptible to manipulations by dietary intake and exercise training [5,11].

Given the crucial role of CHO in sport performance, this review aims to provide an overview of what is known about CHO requirements and recommendations for athletes with an SCI, identify areas of need, and highlight future directions.

## 2. Characteristics of Athletes with a Spinal Cord Injury

The characteristics of athletes with an SCI are variable, and profound differences to AB athletes exist; therefore, a multitude of aspects need to be considered. The scope of motor and sensory impairment stemming from an SCI depends on the level and completeness of the injury, the latter being classified by means of the American Spinal Injury Association (ASIA) Impairment Scale [12,13]. It is noteworthy that this scale omits the assessment of autonomic function, which must be kept in mind when comparing individuals with a para- or tetraplegia [14].

A SCI entails physiological consequences that are relevant to exercise performance and CHO requirements. The magnitude of the injury can vary greatly, and consequently so does the ability to activate the autonomic nervous system (ANS) below the level of lesion [12,15,16,17]. Athletes with a complete SCI rely on their upper body for exercising, which limits their physical work capacity (e.g., reduced peak oxygen consumption, decreased maximal power output and lower mechanical efficiency) [16]. Equally, maximal heart rate might be reduced in individuals injured at or higher than T1-T5 [16,17]. The decreased respiratory function of athletes with an SCI may contribute to the diminished exercise capacity as well [12,16,18]. This respiratory limitation coupled with the impaired ability to redistribute blood flow adequately [16,17] may compromise oxygen and substrate delivery to the active muscles and hence, might attenuate aerobic energy supply [16,19,20]. Furthermore, an impaired vasomotor and sudomotor function from the lesion level downwards causes a blunted thermoregulation that may further burden cardiovascular responses, particularly in hot and humid conditions [15,16,17]. Such conditions may also bear a challenge in terms of energy supply, as higher environmental temperatures increase the dependence on CHO during exercise [21,22].

Individuals with a complete SCI experience severe muscle atrophy below the lesion level that is partly responsible for the reduced energy expenditure during exercise [12,16,23]. Additionally, a myriad of studies found a decline in fat-free mass and an accompanied increase in whole body and regional fat mass in individuals with an SCI (for review see [23]). Conversely, areas above the lesion level seem to remain largely unaffected [23,24]. Likewise, highly trained female wheelchair athletes showed higher fat percentages and a lower fat-free mass in their legs compared to AB controls, while their arms actually revealed a greater fat-free mass [25]. Nevertheless, factors such as gender, lesion level, training status and sport specific variabilities should be taken into account when examining the differences between athletes of various disciplines [26].

Concomitant with the muscle paralysis and the reduced ANS activity, there is a decline in the resting metabolic rate in individuals with an SCI [27,28,29]. Depending on lesion level and completeness, the metabolic rate at rest decreases by 12 to 27% [30] and by typically 25 to 75% [27,29] during exercise in individuals with an SCI compared to AB counterparts. Consequently, total energy expenditure is diminished in athletes with an SCI compared to AB, making a well-balanced nutritional approach even more important in order to maintain a healthy body composition and to achieve the recommended macro- and micronutrient intakes [28,29]. In short, athletes with SCI need to obtain adequate intakes of micro- and macronutrients in the context of lower overall energy intakes. On the other hand, low energy availability (EA) should also be considered as a risk factor in athletes with an SCI, a state that can be acute or chronic [29]. A chronic low EA can have negative health and performance implications and needs to be prevented [29,31,32]. Recent articles suggest an elevated risk for low EA in athletes with an SCI [1,29,33], especially in female athletes [31]. However, further studies are required to specifically investigate the prevalence of low EA in the SCI community, the associated health and performance consequences, and the applicability of cut-off values defined for AB athletes [1,29,31,32,33].

The above-described physiological adaptations implicate numerous secondary complications related to the SCI. Prevalent consequences include spasticity, hepatic dysfunction, immobilization osteoporosis, oxidative stress, chronic systemic inflammation, dyslipidemia and cardiovascular diseases [12,15,23,34,35,36,37]. With that said, the cardiometabolic syndrome occurs frequently in the SCI population, making these individuals susceptible for serious subsequent health issues [15,23,38,39,40]. While there is evidence of reduced cardiovascular disease risk with physical activity in those with an SCI [41], an increased risk in athletes with an SCI [42] has been indicated. A SCI also affects blood glucose regulation. Acquired diseases include glucose intolerance, insulin resistance and type two diabetes mellitus [23,28,43]. Therefore, the ability to utilize CHO is limited [19,44,45], implying glucose oxidation and glycogen synthesis are slightly reduced [45]. Research suggests improvements in glucose and insulin areas under the curve, and increased glucose storage with physical activity in those with an SCI [46]; however, to our knowledge the prevalence or risk of glucose intolerance in athletes with an SCI has not been determined. Since the metabolic flexibility might be decreased as well [45,47,48], the aforementioned metabolic abnormalities may pose a substantial challenge for energy supply. In conclusion, the increased risk or presence of secondary complications and associated metabolic diseases should be considered in the recommendations for the quantity and quality of CHO intakes.

Equally, the gastrointestinal (GI) tract should be an object of special attention in the SCI population. The so-called neurogenic bowel could lead to impaired colonic motility, which might negatively affect nutrient absorption due to increased GI transit times, constipation or upper GI symptoms [49,50,51]. As the composition of the gut microbiota may undergo a change in individuals with an SCI, a state of dysbiosis might further complicate the state [51,52]. Additionally, the potential use of medications to treat secondary health conditions in individuals with an SCI could further worsen the situation by promoting alterations in bioavailability of ingested food, deteriorated gut health and reduced nutrient absorption [1,29,44]. Undesirable, drug-related side effects including nausea, dizziness, suppressed immunity, weight gain and sleep disturbance may negatively impact appetite, diet and the athlete’s daily routine [29,44]. Furthermore, the suboptimal nutritional status reported in individuals with an SCI [28] may also impede proper refueling and recovery, ultimately rising the risk of illness and injury [44]. The end result of the aforementioned GI disturbances would be loss of training days, suboptimal performance or even withdrawal from competition [44].

To summarize, an SCI leads to numerous physiological adaptations, an associated reduced exercise capacity, and serious secondary complications. All of the aforementioned factors significantly affect CHO usage and metabolism in the athletic population with an SCI. As a result, recommendations regarding the quantity and type of CHOs should be personalized to the athlete with an SCI. Individualized CHO recommendations are of major interest to these athletes due to their critical role in supporting optimal health and performance [5,44,53].

## 3. Substrate Metabolism and Utilization during Exercise

Considering the physiological demands of endurance events, CHO and fat represent the main substrates oxidized throughout prolonged endurance exercise [19,22,54,55]. The higher the exercise intensity, the greater the reliance on CHO metabolism with a concomitant downregulation of fat oxidation [7,11,19,22,56]. Other critical determinants are exercise duration, the athlete’s fitness level and nutritional status—i.e., substrate availability [54,55]. The body of an AB endurance athlete (e.g., 65 kg body mass (BM), 12% body fat) stores endogenous CHO as glycogen in the muscles (i.e., 500 g) and the liver (i.e., 110 g) with only 15 g circulating in the blood as glucose [19]. Thus, total endogenous CHO stores in AB individuals account for approximately 10.75 MJ (2560 kcal), indicating their limited capacity and associated vulnerability to exhaustion after exercise of ample length and intensity [19,22]. In contrast, fat stores amount to approximately 7960 g in the above-mentioned athlete, roughly corresponding to 314.16 MJ (74,800 kcal) [19]. Nevertheless, the rate of energy extraction from fat is not sufficient to sustain physical activity of moderate to high intensity [19]. Glycogen and fat storage capacity of individuals with an SCI are difficult to approximate, however, can be estimated individually based on body composition and remaining muscular functionality.

As previously described, individuals with an SCI suffer from profound skeletal muscle atrophy mainly below the level of lesion [23,57], possibly entailing a reduced glycogen storage capacity as compared to AB [1,44,53]. Furthermore, atrophy is followed by a change in fiber type distribution reaching a steady state after up to 70 months post-injury [18,57]. As a result, the paralyzed muscles are predominantly composed of fast type IIx fibers [57]. The opposite holds true above the injury level [18] with a trend towards gradually larger proportions of type I fibers in the deltoid muscle of individuals with an SCI (between 55% and 82%) compared to AB individuals (42%) [58]. Regarding substrate utilization, type I fibers use fat and CHO as fuel sources under aerobic conditions, since they contain more enzymes required for oxidative phosphorylation and fatty acid oxidation [57,59]. In type II fibers, about twice as many glycolytic enzymes are present as in type I fibers, pointing towards a better glycolytic system for energy production [57,58]. However, several studies emphasize the greater dependency of individuals with an SCI on CHO as a fuel substrate over a wide range of exercise intensities [20,60,61,62,63]. Knechtle et al. (2004) found the highest absolute and relative fat oxidation rates at 55% VO_2peak_ in trained hand-cyclists contrary to trained cyclists, who displayed these rates at 75% VO_2peak_ [60]. Additionally, absolute fat and CHO oxidation rates were shown to be lower in athletes with an SCI compared to AB counterparts [60,62]. These findings are probably related to the mode of exercise [60,61,62], the smaller active muscle mass [60,61,62], the fiber type distribution and recruitment pattern [60,61,63], hemodynamic differences [20,60] as well as training status [20,55].

With respect to CHO metabolism, the role of catecholamines should be considered. Once released from the adrenal medulla in response to increasing exercise intensities, epinephrine and norepinephrine promote aerobic glycolysis, glycogenolysis, and gluconeogenesis as well as inhibit insulin-mediated glycogenesis, all of which serve to increase blood glucose levels [19,64]. However, these effects may be attenuated in individuals with an SCI, since the adrenal glands are innervated by the ANS between T5 and T9 [27,63]. Therefore, a higher lesion level connotes lower catecholamine levels [27].

Finally, it should be noted there is a remarkable degree of inter-individual variability existing in substrate utilization [65], which cannot be fully explained by the factors mentioned above nor by gender-related differences [65,66,67].

To demonstrate the metabolic costs evoked by endurance events for individuals with an SCI, a model calculation was conducted (Table 1) using data from Abel et al. (2006) [68]. Thereby, an elite male athlete with a complete SCI at the level of T4 was monitored throughout an outdoor handcycling marathon event [68]. The subject finished the race in 1:48:54 h, which was 20 min slower than his personal best [68]. The slower time resulted from wearing an ergospirometric mask during the race and from stopping due to data collection. Fat and CHO oxidation were quantified according to Jeukendrup et al. (2005), whereby 9.75 kcal/g of fat, 4.07 kcal/g of CHO and a negligible contribution of protein oxidation were assumed (Table 1 and Appendix A) [65]. It has to be mentioned, that this calculation relates to data from one athlete and might not be representative for all athletes with SCI. Nevertheless, the model could be useful in the future to develop CHO guidelines for athletes with SCI.

In order to estimate CHO requirements, the relative contribution of CHO sources was considered, meaning that, at moderate to high intensities, muscle glycogen and plasma glucose deliver 80% and 20% of the required CHO, respectively [65]. This was thought to be applicable to the model, as the athlete managed to maintain on average a performance at 66% VO_2peak_ throughout the race (Table 1) [68]. Concretely, from the total amount of 124.1 g CHO, a contribution of 99.3 g from muscle glycogen and 24.8 g from plasma glucose was calculated (Table 1 and Appendix A). Since the active muscle mass during handcycling is limited to the arms and to parts of the upper body [69], glycogen is assumed to be metabolized from these muscle groups. To approximate the muscle mass of an athlete with an SCI at the level of T4, the study of Flueck (2020) was consulted, wherein the body composition of wheelchair athletes with different lesion levels was described [26]. From these data, the author supplied individual data of an athlete with a T4 lesion level [26] similar to the athlete of Abel et al. (2006) [68]. For the calculation of the available glycogen stores, data from Skrinar et al. (1982) were used [70]. Fat-free mass in the arms was 9.039 kg [26], leading to 150.4 g of stored glycogen determined by the mean glycogen content of the deltoid muscle (e.g., 1.66 g/100 g wet muscle tissue) [70]. By offsetting the upper body muscles associated with handcycling, such as the rotator cuff, the trapezius and pectoralis muscle [69], a 10% higher fat-free mass and consequently glycogen storage capacity were presumed. Therefore, in case of ideal preparations (e.g., filled glycogen stores), the hypothetical athlete had 165.5 g endogenous muscle glycogen accessible throughout the marathon, pointing towards sufficient stores. However, evidence suggests, that glycogen levels below 280–300 mmol per kg dry muscle mass possibly compromises Ca^2+^ release rate from the sarcoplasmic reticulum, thereby diminishing muscle contractility [71,72]. The aforementioned critical threshold equals to 119.4 g glycogen in the calculated model, which would be reached after approximately 52 min without supplementation (assuming a linear glycogen utilization rate of 0.5 mmol/min/kg wet muscle tissue [70]). Hence, 29.3 g/h exogenous CHO supplementation would be required to stay above the critical threshold until the end of the competition. Such exemplary calculations may be fundamental to understanding the actual CHO needs and to develop suitable CHO guidelines for the SCI community. However, the authors acknowledge that this model is only a vague estimation allied with several assumptions. Future work is therefore encouraged to specify the liver’s capacity to perform its task during exercise in individuals with an SCI, as hepatic functions are regulated by the ANS [35,54]. In this context, the lower blood volume of people with a para- or tetraplegia [73] might be essential to consider as well, as it possibly leads to a reduced glucose availability in the blood system.

## 4. Reported Carbohydrate Intake of Athletes with a Spinal Cord Injury

Nutritional status of individuals with a chronic SCI has been examined in previous studies and was summarized in a recently published systematic review [28]. It follows that total daily CHO intake averaged 238 g/day and thus, exceeded minimum recommended values (i.e., 130 g/day) stated in the review [28]. In conclusion, greater energy intakes relative to energy requirements were highlighted in the general SCI population, while identifying multiple nutritional deficiencies [28]. In terms of general health prevention, the intake of sufficient macro- and micronutrients in order to reach the individual requirements is important. In addition, athletes need sufficient energy to support training adaptation and optimize performance [5]. Greater energy intake relative to need is due to several factors including altered body composition (lower fat-free mass), hypometabolic rate, and altered sympathetic nervous system function [74]. While less studied, it is likely that additional barriers including ability to purchase and prepare food contribute to poor dietary intakes. The energy imbalance places these individuals at risk for obesity and associated comorbidities [75]. To date, no guidelines for nutritional intake in individuals with SCI are available [28,44]. 

CHO intake in the athletic SCI population is summarized in Table 2. These studies revealed a relative daily CHO intake that ranges between 2.4 and 7.1 g/kg BM [31,32,76,77,78,79,80,81,82]. As different methods for data collection were used, a comparison between the studies was difficult. However, gender differences in CHO intake are inconclusive. Goosey-Tolfrey and Crosland (2010), Egger and Flueck (2020) as well as Pritchett et al. (2021) demonstrated a 0.9 g/kg BM/day, a 1.6 g/kg BM/day and a 0.4 g/kg BM/day higher intake in male athletes, respectively [31,32,81]. Conversely, Madden et al. (2017) showed equal intakes, whereas Krempien and Barr (2011) declared a 0.5 to 0.8 g/kg BM/day higher intake in female athletes [77,80]. Regarding distinct sport disciplines, a general tendency is difficult to detect, as the majority of participants were wheelchair game players (e.g., basketball, rugby and tennis). Interestingly, great variability and large intake ranges were reported even within the same sport [79]. Additionally, no study subdivided the results into different type of sports. The same holds true for the level of injury. Only Gerrish et al. (2017) analyzed the findings separately for athletes with para- and tetraplegia and revealed a statistically insignificant, lower relative daily CHO intake in those with tetraplegia [76]. The time point of the season and the training volume on the days investigated appear to influence the CHO intake as well [76,78,79].

Relative daily CHO intake of sedentary individuals with an SCI ranged from 1.6 to 3.4 g/kg BM [83,84,85,86,87]. Considering the fact, that the athletic population with SCI is presumed to have higher activity levels and therefore higher energy needs [81], a greater difference to the range outlined above might be expected. Thus, these data may either indicate an excessive CHO intake in the general population with SCI or a rather low CHO intake in athletes with an SCI. The former assumption is corroborated by the high prevalence of obesity and obesity-related disorders in sedentary individuals with an SCI [23] as well as the increased energy intake related to energy expenditure [28]. The second hypothesis is supported by evidence that suggests low EA [31] and poor nutritional knowledge in athletes with an SCI, especially in topics related to sports nutrition [88]. Consequently, periodic proper nutritional education in both the general and athletic population with SCI might help (i) to ensure a diet full of nutrient-dense food and high quality choices [29,80]; (ii) to minimize the risk of nutritional inadequacies [29,44,79]; (iii) to improve health and performance by calculating specific needs on an individual basis [31,44,76,78,80].

In summary, it can be stated that relative daily CHO intake seems to be at the lower limit of current guidelines (3 to 12 g/kg BM/d) for AB athletes [5,76,77,78,79,80,81]. Moderate and high volume and/or intensity training may warrant a diet comprising greater amounts of CHO [5,6]. Sanz-Quinto et al. (2019) investigating an elite wheelchair marathoner reported a relative daily CHO intake that was similar to the AB recommendations [5,82]. However, applying the same nutritional standards in athletes with an SCI as in AB athletes is probably not reasonable [77,78,80,81]. Therefore, drawing a conclusion by solely having AB recommendations as a reference may not be appropriate [76], particularly in view of the unique metabolic and physiological changes affecting individuals with an SCI. Moreover, the paucity of research and the low number of subjects recruited for the available studies (see Table 2) may not truly reflect the whole athletic population with SCI. As a result, the aforementioned interpretations are only preliminary.

## 5. Carbohydrate Supplementation in Athletes with a Spinal Cord Injury

According to current research in AB athletes, CHO supplementation is most likely associated with performance benefits due to a variety of mechanisms, such as muscle and liver glycogen sparing, delivery of an exogenous muscle substrate, preserved plasma glucose levels and CHO oxidation rates as well as the activation of central nervous system effects [5,8,55]. However, only little is known about CHO supplementation before and during exercise in the SCI community (Table 3).

Jung and Yamasaki (2009) investigated the effect of pre-exercise CHO ingestion on fat and CHO oxidation during prolonged arm cranking exercise [93]. Whereas AB subjects showed a significantly higher CHO oxidation in the CHO compared to the placebo (PLA) trial, no effect of glucose ingestion on CHO oxidation was found in individuals with paraplegia [93]. In addition, the latter showed higher fat oxidation rates in the CHO and PLA trial compared to the AB participants, although the significant difference was only observed in the CHO trial [93]. The authors stated that the different fiber type distribution as well as the dependence on a wheelchair in daily life may possibly explain their findings [93].

Two studies evaluated the influence of pre-exercise CHO ingestion compared to PLA on performance in individuals with an SCI [89,94], with one study continuing supplementation to some extent during exercise [94]. Temesi et al. (2010) divided six athletic individuals with a thoracic or cervical level of injury into two groups and supplemented them with either a CHO or PLA drink in a randomized, counter-balanced order [94]. CHO ingestion failed to improve performance in the all-out time trial [94]. This outcome contradicts the finding of Spendiff and Campbell (2003), since they demonstrated a significantly enhanced all-out test performance after ingesting a CHO drink compared to the PLA drink [89]. Reasons for the discrepancies that occurred between the two studies may include different dosages (absolute vs. relative) and time points of ingestion, various characteristics of injury (e.g., type, level and completeness) and unequal fitness levels. Additionally, Temesi et al. (2010) led the participants exercising in a fasted state [94], while the subjects of Spendiff and Campbell (2003) were three hours postprandial [89]. Both studies displayed performance variabilities and seemed to have a low statistical power, as evidenced by the small sample size, therefore, a final conclusion cannot be made.

Another work of Spendiff and Campbell (2005) analyzed the effect of two different glucose concentrations consumed before 65% VO_2peak_ exercise on subsequent all-out test performance in wheelchair athletes [91]. Individuals with paraplegia either received a high (11%) or low (4%) CHO dosage 20 min prior to 60 min 65% VO_2peak_ exercise in a randomized, double-blind, crossover design [91]. No differences in subsequent all-out test performance were found between the high and the low CHO trial [91]. Nevertheless, the tendency for higher blood glucose concentrations and respiratory exchange ratios throughout the high CHO trial were probably indicative of increased CHO oxidation rates [91]. Moreover, the significantly lower free fatty acid concentrations at the end of the high CHO trial might be equally reflective of the greater shift in substrate utilization from free fatty acids to CHO [91]. Thus, the higher distances covered and power outputs achieved during the high CHO trial may be clinically relevant, albeit not statistically significant (*p* = 0.08) [91]. For these reasons, the authors argued for the supplementation with the higher glucose concentration to possibly be the better choice for athletes with an SCI [91]. A comparison to the studies above is difficult due to the lack of a control group.

Only two studies focused on the examination of CHO supplementation during exercise in athletes with an SCI [90,92]. The first study applied two different drinking schedules in a randomized, crossover design: (1) four doses of 162 mL of a 7.6% CHO beverage, provided at the start and after 20, 40 and 60 min of exercise at 65% VO_2peak_ or (2) two doses of 324 mL of the same 7.6% CHO beverage, but provided at the start and after 60 min of exercise [90]. The distance covered in the subsequent all-out performance test showed no significant difference between the two drinking schedules [90]. Still, the authors emphasized the greater distance covered in the 20 min time trial by each athlete in the trial where they ingested four doses of 162 mL 7.6% CHO beverage and underlined the possible impact of this 11% improvement on real world athletic performance [90]. Besides the slightly higher increase in exercise intensity during the all-out test (86% vs. 81% VO_2peak_), the more frequent drinking schedule also led to sustained blood glucose concentrations during and significantly lowered free fatty acid concentrations after the one hour 65% VO_2peak_ exercise [90]. Thus, the ingestion of recurrent small doses (162 mL containing 7.6% CHO) of a 7.6% CHO beverage throughout wheelchair exercise was regarded as the best strategy for athletes with low lesion paraplegia [90]. Building on these findings and the limitations mentioned by Spendiff and Campbell (2004) [90], the second study administered four doses (200 mL each) of either a 8% CHO or PLA drink during exercise and investigated its effect on all-out test performance in wheelchair athletes [92]. The CHO and PLA beverages were given at timed intervals (15, 30, 45, 60 min) during a one hour 65% VO_2peak_ exercise, which was followed by a 30 min all-out test [92]. The CHO trial resulted in significantly higher blood glucose concentrations, CHO oxidation rates, and respiratory exchange ratios during the latter stages of the performance test [92]. However, again, CHO supplementation revealed no superior all-out test performance [92]. Indeed, six out of eight participants covered a greater distance during the PLA as against the CHO trial [92]. A comparison between the two studies is rather difficult, since the second study worked with a PLA control group and a fixed pre-exercise meal (2 g CHO/kg BM) ingested three hours before exercise [92], while the first one did not [90]. Moreover, other methodological disparities, such as the length of the performance trial (20 vs. 30 min), the volume of beverage and amount of CHO supplemented as well as unequal fitness levels of participants further complicate the comparison. Consequently, research examining the ideal timing and CHO dosage in athletes with an SCI has yet to be further investigated, whereby larger sample sizes should be sought in order to raise statistical power. In conclusion, key differences between the studies as well as the scarce literature available makes it virtually impossible to determine the effectiveness of CHO supplementation before and during exercise in the SCI population at present. Nevertheless, CHO supplementation during exercise seemed to increase CHO oxidation rate and stabilize blood glucose concentration. Even though, results for performance outcome parameters such as time trial performance are inconclusive in a statistical perspective. A clear tendency for a performance benefit from CHO intake during endurance exercise can be observed. This could be even more important during longer events (e.g., wheelchair marathon racing, handcycling) as the studies only used 20 to 30 min time trial durations.

## 6. Recommendations for Athletes with a Spinal Cord Injury

To date, there are no specific CHO guidelines for athletes with an SCI [1]. Instead, current recommendations are limited to those applied in AB athletes [5,95], most notably because of the limited research conducted in SCI [1]. However, by reflecting on all the aspects discussed in this review, sufficient metabolic and physiological differences became apparent to substantiate the need for properly adjusted CHO recommendations for the SCI community.

Irrespective of any neurological impairment, adequate CHO availability is a cornerstone of an athlete’s performance capacity [5]. Nutritional strategies to support CHO availability should be implemented before, during, and after competitions or high quality and intensity training sessions—as suggested in AB athletes [5]. In general, CHO requirements depend on a variety of factors including the frequency, intensity, duration, and type of exercise, the athlete’s goal, and the environment [5]. Therefore, the quantity and timing of CHO ingestion are of major importance. Within the former aspect, value should be attached to a balanced nutritional approach. The paradigm of periodized nutrition [5,55,95,96] may represent a guide for athletes with an SCI to base their fueling needs on training duration, intensity, and total volume [31,96]. In view of their reduced energy expenditure, such a periodized approach could enable these athletes to ensure optimal training adaptations and recovery without restricting their diet and thus, support maximal performance capacity [31]. Work is required to ensure appropriate EA despite lower energy budgets. Similarly, the time point of ingestion deserves closer attention. While playing a minor role in covering the daily needs, it should be considered in pre-exercise nutritional preparations, specifically with respect to dyspepsia. As previously mentioned, gastric emptying can be affected in individuals with an SCI, which might delay CHO absorption and availability for oxidation [1]. Hence, it is essential to allow for sufficient digestion time before exercise in order to ensure filled endogenous glycogen stores and to minimize the risk of GI problems occurring during the event [1]. It might be equally relevant to consider GI transit times during exercise, as physical activity itself and the tight position in a sports wheelchair may prolong them [1]. Thus, a carefully planned pre-competition meal should probably gain even more importance. This might be further underpinned by the fact that drinking throughout wheelchair exercise disrupts the rhythm and momentum of the athlete [90]. Therefore, athletes omit drinking too often. Conversely, the smaller amount of metabolically active muscle mass may point towards a lower glycogen storage capacity [1,44,61]. Accordingly, glycogen depletion possibly occurs earlier [1,44], most notably in light of the marked reliance on CHO as an exercise fuel substrate. This would increase the necessity of exogenous CHO supplementation during exercise–comparable to the practices currently applied in AB athletes. Briefly, AB athletes are advised to ingest 30 to 60 g CHO/h of a 6 to 8% CHO solution throughout endurance exercise lasting 60 to 150 min [5,66,95]. The same holds true for stop and go sports of equal duration [5,95]. Greatest intakes are prescribed when performing ultra-endurance exercise (>150 min) with amounts up to 90 g CHO/h, whereby co-ingestion of glucose and fructose in a 2:1 ratio is highly advocated [5,95]. It is still to be clarified whether or not these quantities would also be applicable to athletes with an SCI. Thus, future work is encouraged to examine the optimal CHO quantity, dosage and timing in athletes with an SCI.

Regarding delayed GI transit times, dietary fibers might produce relief [53]. Individuals with an SCI are recommended to ingest 25 to 30 g fibers every day, as fibers could help improve GI problems, the gut microbiome, the quality of a diet, and health in general [53,97]. In addition to the quantity of dietary fibers, the type requires consideration, as there are differing effects on health and the GI tract depending of fiber classification [98]. It is also noteworthy that high-fiber diets may influence bowel function in individuals with an SCI contrary to expectations [53]. Therefore, the low daily dietary fiber intake reported in athletes with an SCI (i.e., 11.4 to 22.0 g/day) [32,77,80,81] is not surprising and is likely justifiable by the potential consequences of high-fiber diets on individual bowel management routines [1,28,53,81]. The timing of dietary fiber in relation to exercise will also need to be considered, as dietary fiber in the pre-exercise meal in AB athlete is associated with increased exercise-induced GI symptoms [5,99].

The large heterogeneity among athletes with an SCI poses one of the major challenges to the establishment of scientifically supported CHO guidelines [28]. The lesion level and the completeness of the injury affect the amount of active muscle mass. Therefore, both factors coupled with the ability to activate the ANS do not only critically affect the quantity and timing of ingestion, but also the frequency, intensity, duration as well as type of exercise feasible for an athlete with an SCI. As a result, energy expenditure and nutritional requirements are highly dependent on those important determinants. Furthermore, even within the same categorized lesion level, physiological characteristics between individuals with an SCI are highly diverse [92], ultimately representing a further barrier to the determination of CHO needs. Additionally, certain circumstances and environmental conditions, such as physical injuries, medications and exposure to cold or heat, temporarily increased energy requirements, while others (e.g., lower exercise volume, decrease in fat-free mass) reduce them [5]. Consequently, the need to scale the AB recommendations to the SCI community is accentuated by the various characteristics uniquely affecting athletes with a para- or tetraplegia.

Finally, it is worth mentioning that current CHO guidelines for the daily energy needs, the pre-event preparation phase, and the recovery period of AB athletes are provided per kg of BM as a proxy for the volume of exercising muscles and muscle stores [5,95]. With that said, the direct transfer of nutritional guidelines from AB athletes to athletes with an SCI seems once again not reasonable. Therefore, establishing specialized recommendations that are scaled to the fat-free mass of individuals with an SCI might demonstrate a plausible approach, which should be thoroughly analyzed in prospective studies. Present CHO guidelines for AB athletes during exercise–as outlined above–are given in absolute amounts, because evidence underlines little difference in the oxidation of exogenous CHO in proportion to BM [88]. Nevertheless, they should probably be adjusted to the actual energy requirements of athletes with an SCI, while mouth rinsing as well as co-ingestion of glucose and fructose may also constitute points for discussion. However, it must be admitted that all studies investigating CHO intake and/or supplementation in athletes with an SCI are confined by a small sample size, leading to limited data for interpretations. Thus, future research is urgently required to allow the development of scientifically based CHO recommendations for intake and supplementation in the athletic population with SCI. Ultimately, researchers and practitioners may need to approach the concept of recommendations from a different angle for these athletes. It seems that a series of factors have to be considered and adjusted recommendations need to be provided rather than absolute ranges, as is how guidelines of AB athletes are presented.

## 7. Future Directions

Going forward, we should seek for the recruitment of large, homogenous samples (e.g., multicenter studies) in order to overcome the low statistical power of the present literature, as well as the inherent heterogeneity among individuals with an SCI. The implementation of multicenter studies may help to address these issues [31]. In compliance with the findings of this review, the following research priorities are identified:Analysis of daily CHO intake as well as pre-, during and post-exercise CHO supplementation to represent a larger range of the athletic community with SCI. For instance, prospective research should not only focus on wheelchair game players, but also include other type of sports (e.g., endurance or winter sports) to broaden the knowledge about current practices.Evaluation of energy requirements for athletes with various lesion levels and especially within diverse sport disciplines. The clarification of energy costs for the athletic population with an SCI would provide the basis for understanding the energy needs of such athletes. This would aid in establishing sound CHO recommendations.Characterization of gender differences related to daily energy demands as well as substrate utilization during exercise. For example, evidence revealed a lower reliance on CHO metabolism throughout endurance exercise in AB women compared to AB men, most likely through hormonal differences [67]. This calls for studies involving more female individuals with para- or tetraplegia than currently exists.Assessment of glycogen storage capacity and utilization rate in athletes with an SCI. This would facilitate the definition of ideal CHO fueling and supplementation strategies.Evaluation of the autonomic dysfunction and its effect on CHO metabolism. Along with different hormones, the ANS is known to be involved in regulating exercise metabolism [19,27]. However, to what extent an altered sympathetic innervation of the adrenal medulla, liver and pancreas—to name but a few—caused by the SCI [35,94] may influence substrate mobilization and with that CHO metabolism should be further elucidated (e.g., hepatic glycogen storage, glycogenolysis and gluconeogenesis).

## 8. Conclusions

In conclusion, the ever-growing popularity of Paralympic sports combined with the characteristics uniquely affecting athletes with an SCI make the case for specialized nutritional recommendations. The determination of CHO recommendations will be a significant challenge, as dietary CHO intake and supplementation both need to be considered. Furthermore, CHO recommendations will need to consider optimal EA and other macronutrient requirements in the context of lower energy budgets, to allow for ideal training adaptations and peak performances. The large heterogeneity among the athletic population with SCI resulting from the metabolic and physiological adaptations further confuse the issue. Owing to the paucity of research and the low statistical power of studies conducted with the SCI community, it is currently not feasible to provide scientifically based recommendations for CHO intake and supplementation. Prospective studies should, therefore, not only analyze proper daily CHO needs, but also assess the effectiveness of pre-, during and post-exercise CHO supplementation. In the absence of sufficient high-quality evidence, tailoring nutritional strategies to the individual athlete seems to represent a valuable approach in order to meet unique performance goals and health optimization This calls for personalized nutritional counseling and education from nutrition professionals with a performance background, ideally a certified sports dietitian. Within this support, ideal CHO quantity, quality and timing should be specifically evaluated in conjunction with individual preferences, tolerances, and energy requirements. This implies that CHO fueling and supplementation protocols need to be well-tested through intense training sessions and cycles to ensure their effectiveness, especially regarding minimizing GI distress. Such a personalized counseling will make the best of each athlete’s individual capabilities—an aspect that gains even more importance when it comes to medals, since differences between winning and losing are often remarkably small in real-world athletic performance. Given the importance of CHO in exercise performance and health, establishing unique CHO guidelines for athletes with an SCI is of ultimate importance.

## Figures and Tables

**Table 1 nutrients-13-02177-t001:** Model calculation of the metabolic costs associated with performing a handbike marathon.

Specific Parameters	Results	References
Time to complete [min]	108.9	Parameters given by Abel et al. [68]
Mean power [W]	84.1	
Gross efficiency [%]	15.3	
Mean VO_2_ [%VO_2peak_]	66.4	
Energy expenditure [kcal/min]	7.7	
Energy expenditure [kcal/h]	462	
Total energy cost [kcal]	838	
Fat oxidation [g/min]	0.31	Parameters calculated according to Jeukendrup et al. [65]
Fat mass oxidized [g]	34.1	
Energy delivered by fat [kcal]	333	
Proportion of fat oxidation [%]	40	
CHO oxidation [g/min]	1.14	
CHO mass oxidized [g]	124.1	
Energy delivered by CHO [kcal]	505	
Proportion of CHO oxidation [%]	60	

CHO = carbohydrate; VO_2_ = oxygen consumption; VO_2peak_ = peak oxygen consumption.

**Table 2 nutrients-13-02177-t002:** CHO intake in athletes with an SCI.

Author	Subjects	Methods	Daily CHO Intake
Goosey-Tolfrey and Crosland (2010) [81]	14 W 9 M elite wheelchair athletes (basketball and tennis)	7-day weighed food diary over 7 consecutive days during preparation phase	W: 3.40 ± 0.71 g/kg/d M: 4.31 ± 1.46 g/kg/d
Krempien and Barr (2011) [77]	8 W 24 M elite athletes with an SCI (rugby, basketball, alpine skiing and athletics)	3-day self-reported food journal kept at home and during training camp	Training camp: W: 4.9 ± 1.1 g/kg/d M: 4.4 ± 1.2 g/kg/d Home: W: 4.9 ± 1.4 g/kg/d M: 4.1 ± 1.5 g/kg/d
Grams et al. (2016) [79]	17 M elite wheelchair basketball players	3-day weighed food journal over 3 consecutive days during 3 training camps over 2 consecutive years	Average: 3.9 [1.8;8.1] g/kg/d Training camp 1: 3.1 [2.8;7.1] g/kg/d Training camp 2: 3.5 [1.8;8.1] g/kg/d Training camp 3: 4.9 [2.7;7.4] g/kg/d
Ferro et al. (2017) [78]	11 M elite wheelchair basketball players	3-day weighed food diary in 2 months during precompetitive period	May: 3.76 ± 1.30 g/kg/d June: 4.24 ± 1.92 g/kg/d
Gerrish et al. (2017) [76]	19 W 20 M Canadian and US elite wheelchair athletes (tennis, track, basketball, and rugby)	Self-reported, single 24 h food journal in autumn and winter training camps	Autumn: W: 3.6 ± 0.8 g/kg/d M: 3.8 ± 0.9 g/kg/d Winter: W: 3.4 ± 1.0 g/kg/d M: 2.8 ± 0.5 g/kg/d
Madden et al. (2017) [80]	22 W 18 M elite wheelchair athletes (different sport disciplines)	3-day self-reported food journal	W: 3.5 ± 1.0 g/kg/d M: 3.5 ± 1.2 g/kg/d
Sanz-Quinto et al. (2019) [82]	1 M elite wheelchair marathoner	Individualized nutrition program over 7 weeks, self-reported nutritional diary to record daily intake	Pre-altitude: 7.10 ± 1.20 g/kg/d Post-altitude: 6.30 ± 0.80 g/kg/d
Egger and Flueck (2020) [31]	6 W 8 M Swiss elite wheelchair athletes (different sport disciplines)	7-day weighed food diary during pre-season	W: 2.4 ± 0.7 g/kg/d M: 4.0 ± 1.0 g/kg/d
Pritchett et al. (2021) [32]	9 W 9 M Canadian and US para-athletes (wheelchair track/marathon and basketball)	7-day food log over 7 consecutive days	W: 3.7 ± 0.8 g/kg/d M: 4.1 ± 1.3 g/kg/d

Daily CHO intake is given in g per kg body mass per day. CHO = carbohydrate; M = men; SCI = spinal cord injury; W = women.

**Table 3 nutrients-13-02177-t003:** CHO supplementation in individuals with an SCI.

Author	Subjects	Exercise	Supplementation	Results (Mean ± SD)
Spendiff and Campbell (2003) [89]	8 M with SCI (3 with P, 5 with spina bifida)	60 min arm cranking at 65% VO_2peak_, followed by 20 min all-out test	600 mL of either a CHO (48 g of dextrose monohydrate + sugarless orange) or a PLA drink (artificially sweetened, orange flavored) 20 min before 65% VO_2peak_ exercise	↑ all-out test performance in CHO vs. PLA trial (*p* = 0.05) PLA: 10.2 ± 1.0 km CHO: 10.8 ± 0.7 km
Spendiff and Campbell (2004) [90]	7 M with SCI (2 with P, 5 with spina bifida)	60 min wheelchair ergometry at 65% VO_2peak_, followed by 20 min all-out test	648 mL of a CHO (48 g of dextrose monohydrate + sugarless orange) administered in two different drinking schedules (4 × 162 mL or 2 × 324 mL) during 65% VO_2peak_ exercise	↔ all-out test performance in both drinking schedules (*p* > 0.05) 162 mL: 5.5 ± 2.1 km 324 mL: 4.9 ± 1.2 km
Spendiff and Campbell (2005) [91]	8 M with SCI (3 with P, 5 with spina bifida)	60 min wheelchair ergometer exercise at 65% VO_2peak_, followed by 20 min all-out test	600 mL of either a high CHO drink (72 g dextrose monohydrate + no-added sugar cordial) or a low CHO drink (24 g dextrose monohydrate + no-added sugar cordial) 20 min before 65% VO_2peak_ exercise	↔ all-out test performance in high vs. low trial (*p* > 0.05) low: 5.0 ± 1.5 km high: 5.2 ± 1.6 km
Hynes (2009) [92]	8 wheelchair athletes (6 M, 2 F; 2 with T, 5 with P; 1 with CP)	60 min arm cranking at 65% VO_2peak_, followed by 30 min all-out test	4 × 200 mL of either a CHO (in total 42.2 g sucrose + 21.1 g glucose) or a PLA (1.6 g sucrose + non-caloric sweetener) drink during 65% VO_2peak_ exercise	↔ all-out test performance in CHO vs. PLA trial (*p* > 0.05) PLA: 5.3 ± 1.3 km CHO: 5.3 ± 1.2 km
Jung and Yamasaki (2009) [93]	6 M with P 7 AB M	60 min arm cranking at 80% of individual lactate threshold	500 mL of either a glucose solution (1 g/kg BM, CHO) or plain water immediately before exercise	↑ fat oxidation in P vs. AB (*p* < 0.05) in CHO trial
Temesi et al. (2010) [94]	6 individuals with complete SCI (5 M, 1 F; 3 with T, 3 with P)	60 min arm cranking at 65% VO_2peak_, followed by 20 min all-out time trial	125 mL of either a CHO drink (lime cordial and artificial sweetener + 0.5 g/kg BM maltodextrin) or a PLA drink (lime cordial + artificial sweetener) every 15 min from 15 min before and up to 30 min of 65% VO_2peak_ exercise	↔ time trial performance in CHO vs. PLA trial (*p* > 0.05) PLA: 51.8 ± 22.3 kJ CHO: 54.2 ± 22.6 kJ

AB = able-bodied; BM = body mass; CHO = carbohydrate; CP = cerebral palsy; M = men; F = female; PLA = placebo; P = paraplegia; SCI = spinal cord injury; SD = standard deviation; T = tetraplegia; VO_2peak_ = peak oxygen consumption; W = women; ↑ means a significant increase; ↔ means no difference

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
