# Peer review of "Carbohydrate Considerations for Athletes with a Spinal Cord Injury"

_nutrients, 2021, doi:10.3390/nu13072177_

Round 1

Reviewer 1 Report

Thank you for the opportunity to review this important manuscript, and apologies for the delay in submitting my review.

I felt that this was a comprehensive overview of the literature to date and practical considerations regarding carbohydrate recommendations for athletes with spinal cord injury.

It was generally well written. The language used was highly academic which may be a barrier for some readers including clinicians, albeit appropriate for a scientific journal publication.

The authors presented a well-developed introduction providing a comprehensive overview of relevant literature and presenting a compelling rationale for this paper. They went on to outline key considerations, a detailed overview of relevant literature, worked examples of the need to modify CHO recommendations for SCI, and thoughtful recommendations for practice and future research.

I have outlined a few suggestions that would enhance readability of the manuscript:

Line 76 – it would be helpful to explain the term ‘autonomic completeness’ for readers less familiar with SCI

In Section 3: Substrate Metabolism and Utilization during Exercise, the authors present a useful model that may support the development of CHO guidelines for SCI athletes. However, it would be helpful for the authors to make explicitly clear that the findings presented relate to the specific characteristics/circumstances of one hypothetical SCI athlete, and that the full process should be undertaken for individual athletes. For example, it could be made more clear that the levels of glycogen storage and utilization relate to this particular athlete, and are not representative of all SCI athletes.

 Line 260 – it would be helpful to expand with a further sentence noting why it is common for energy intake to exceed energy requirements, and the potential impacts for health and performance.

Lines 404-405 – it would be helpful here to recap on the key findings related to metabolic and physiological differences

Line 431-432 – “This might be further underpinned by the fact that several wheelchair athletes face the difficulty of drinking throughout exercise” – This line could use clarification. Do you mean that several athletes have reported difficulty drinking through exercise? Or that there are certain sports where this is problematic?

Line 494 – I suggest editing ‘supreme importance’ to a term that is less emotive

Author Response

Line 76 – it would be helpful to explain the term ‘autonomic completeness’ for readers less familiar with SCI

Thank you for this comment. We changed the wording to make it clearer.

In Section 3: Substrate Metabolism and Utilization during Exercise, the authors present a useful model that may support the development of CHO guidelines for SCI athletes. However, it would be helpful for the authors to make explicitly clear that the findings presented relate to the specific characteristics/circumstances of one hypothetical SCI athlete, and that the full process should be undertaken for individual athletes. For example, it could be made more clear that the levels of glycogen storage and utilization relate to this particular athlete, and are not representative of all SCI athletes.

Thank you for this comment. We have added two sentences to make it more clear, that the calculation is hypothetical and based on an individual athlete. We fully agree, that this is not a calculation which is representative for all athletes with SCI but it might be useful in the future for determination of CHO guidelines.

 Line 260 – it would be helpful to expand with a further sentence noting why it is common for energy intake to exceed energy requirements, and the potential impacts for health and performance.

 We agree with your comment and expanded the section.

Lines 404-405 – it would be helpful here to recap on the key findings related to metabolic and physiological differences

We added a sentence to recap the key findings of the studies.

Line 431-432 – “This might be further underpinned by the fact that several wheelchair athletes face the difficulty of drinking throughout exercise” – This line could use clarification. Do you mean that several athletes have reported difficulty drinking through exercise? Or that there are certain sports where this is problematic?

This sentence has been clarified to make sure, that drinking during exercise disrupts the rhythm of the exercise and therefore, athlete omit drinking too often.

Line 494 – I suggest editing ‘supreme importance’ to a term that is less emotive

This has been changed.

Reviewer 2 Report

Thank you for the opportunity to review your manuscript. I’d like to congratulate the authors on a well written manuscript on an topic which is certainly in need of more research and attention. I think the authors do a very good job of explaining why specialised recommendations are required with a clear overview of the characteristics of SCI physiology, current review on CHO intake, and future research recommendations. I have some minor comments which I’ve broken down by section. For the most part my comments are recommendations, which if the authors feel are inappropriate please don’t feel obliged to, but please provide a rationale as to why.

  1. Introduction

Concisely written and sets the scene nicely.

  1. Characteristics of Athletes with a spinal cord injury

Line 109 – ‘a well-balanced nutrition even more important’. This sentence doesn’t quite make sense in its current form, please amend – I suggest ‘making a well-balanced nutritional approach…..’ Secondly, more important for what? I believe the authors are referring to the maintenance of body composition – please make clear for the reader.

Line 117 – Regarding EA cut off values - excellent point.

Line 125 – Is ‘hazards’ the most applicable term here? Forgive my naivety if it is.

  1. Substrate Metabolism and Utilization during Exercise

Line 185 – I think the line ‘type I fibers use fat as major fuel source’ is misleading. As the authors mention oxidative phosphorylation is prevalent in these fibres and that includes from CHOs. If the authors wish to retain this then please explain more clearly to the reader as I worry it could be misunderstood that Type I fibres are reliant only on fat. It’s also not clear within these sentences if this is specific to SCI or a general point.

Line 212 – The mention that the athlete beat his PB by 20mins, is there a reason for this inclusion? I’d mention that this was on a treadmill and perhaps the reason for such an improvement. Perhaps better to compare to times to those in competitive marathons?

General comment – Is it perhaps worth pointing out that this model is only applicable for certain events?

  1. Reported Carbohydrate Intake of Athletes with a Spinal Cord Injury

A concise well written section. My only thought is that perhaps at the start of the section I’d be tempted to point out to the reader there are no current SCI CHO guidelines, I’m aware at this comes within it’s own section later in the manuscript but you could mention that. Perhaps also worth pointing out the discrepancies in methods used to collect the dietary data and the issues that come with that.

  1. Carbohydrate Supplementation in Athletes with a Spinal Cord Injury

Table 3 – I think a column which includes the study design would be useful to the reader, but not a requirement. A suggestion rather than a request.

Line 367 – when discussing the doses please include the CHO solution or in grams

Line 378 – please define what is considered a ‘small dose’

Line 382 – please include the CHO solution or in grams

  1. Recommendations for Athletes with a Spinal Cord Injury

Line 408 – ‘Irrespective of the neurological impairment’ should this be ‘of any neurological…’ ?

Line 411 – I’d remove ‘just like it is’ and replace with ‘as suggested…’

Line 415 – ‘ to a balanced nutrition’ perhaps is me but this doesn’t make sense. I think it’s missing a word, I suggest adding ‘approach’ to the end.

Line 446 – ‘SCI were recommended’ should this not be ‘are’?

Line 451 – I would add ‘reported’ to ‘the low daily’

Line 475 – Excellent point and following section.

  1. Future Directions

Excellent concise recommendations that I would agree with based on the information contained within this manuscript.

  1. Conclusions

A well written section which nicely concludes the manuscript.

Author Response

  1. Introduction

Concisely written and sets the scene nicely.

  1. Characteristics of Athletes with a spinal cord injury

Line 109 – ‘a well-balanced nutrition even more important’. This sentence doesn’t quite make sense in its current form, please amend – I suggest ‘making a well-balanced nutritional approach…..’ Secondly, more important for what? I believe the authors are referring to the maintenance of body composition – please make clear for the reader.

This has been clarified.

Line 117 – Regarding EA cut off values - excellent point.

Thank you very much!

Line 125 – Is ‘hazards’ the most applicable term here? Forgive my naivety if it is.

This has been changed.

  1. Substrate Metabolism and Utilization during Exercise

Line 185 – I think the line ‘type I fibers use fat as major fuel source’ is misleading. As the authors mention oxidative phosphorylation is prevalent in these fibres and that includes from CHOs. If the authors wish to retain this then please explain more clearly to the reader as I worry it could be misunderstood that Type I fibres are reliant only on fat. It’s also not clear within these sentences if this is specific to SCI or a general point.

We fully agree with your comment and added CHO as a fuel source. This is a general comment not specifically for SCI.

Line 212 – The mention that the athlete beat his PB by 20mins, is there a reason for this inclusion? I’d mention that this was on a treadmill and perhaps the reason for such an improvement. Perhaps better to compare to times to those in competitive marathons?

General comment – Is it perhaps worth pointing out that this model is only applicable for certain events?

Thank you for this comment. This has been clarified in the article as the race was an outdoor race under real life conditions wearing an ergospirometric mask. The athlete was 20 min slower in the race compared to his personal best. Furthermore, we added, that this calculation is only a rough estimation and is not representative for all athletes.

  1. Reported Carbohydrate Intake of Athletes with a Spinal Cord Injury

A concise well written section. My only thought is that perhaps at the start of the section I’d be tempted to point out to the reader there are no current SCI CHO guidelines, I’m aware at this comes within it’s own section later in the manuscript but you could mention that. Perhaps also worth pointing out the discrepancies in methods used to collect the dietary data and the issues that come with that.

We are grateful for this comment and included both aspects in the section.

  1. Carbohydrate Supplementation in Athletes with a Spinal Cord Injury

Table 3 – I think a column which includes the study design would be useful to the reader, but not a requirement. A suggestion rather than a request.

Thank you very much for this comment. We tried to divide the study design into the exercise task and the supplementation. As the table provides already a lot of information, we do not want to make it more complicate to read it by adding another column or by putting the exercise task and the supplementation into one column.

Line 367 – when discussing the doses please include the CHO solution or in grams

Thank you for this comment. We added % of the solution as this seems more appropriate than grams as the volume was different in the different protocols.

Line 378 – please define what is considered a ‘small dose’

This has been added.

Line 382 – please include the CHO solution or in grams

This has been added.

  1. Recommendations for Athletes with a Spinal Cord Injury

Line 408 – ‘Irrespective of the neurological impairment’ should this be ‘of any neurological…’ ?

Yes of course, this has been changed.

Line 411 – I’d remove ‘just like it is’ and replace with ‘as suggested…’

This has been changed.

Line 415 – ‘ to a balanced nutrition’ perhaps is me but this doesn’t make sense. I think it’s missing a word, I suggest adding ‘approach’ to the end.

This has been changed.

Line 446 – ‘SCI were recommended’ should this not be ‘are’?

We agree.

Line 451 – I would add ‘reported’ to ‘the low daily’

This has been added.

Line 475 – Excellent point and following section.

Thank you for this comment.